# Sensitivity Analysis of the Rainfall–Runoff Modeling Parameters in Data-Scarce Urban Catchment

**Héctor A. Ballinas-González** [1,*] **, Víctor H. Alcocer-Yamanaka** [2] **, Javier J. Canto-Rios** [3] **and Roel Simuta-Champo** [1]

[1] Mexican Institute of Water Technology, Paseo Cuauhnáhuac 8532, Progreso, Jiutepec, Morelos 62550, Mexico; roel_simuta@tlaloc.imta.mx

[2] National Water Commission, Insurgentes Sur 2416, Copilco El Bajo, Coyoacan, CDMX 04340, Mexico; yamanaka@conagua.gob.mx

[3] Autonomous University of Yucatán, Industrias no contaminantes S/N, Mérida, Yucatán 150, Mexico; javierj.canto@correo.uady.mx

* Correspondence: hballinas@tlaloc.imta.mx

**Abstract:** Rainfall–runoff phenomena are among the main processes within the hydrological cycle. In urban zones, the increases in imperviousness cause increased runoff, originating floods. It is fundamental to know the sensitivity of parameters in the modeling of an urban basin, which makes the calibration process more efficient by allowing one to focus only on the parameters for which the modeling results are sensitive. This research presents a formal sensitivity analysis of hydrological and hydraulic parameters—absolute–relative, relative–absolute, relative–relative sensitivity and $R^2$—applied to an urban basin. The urban basin of Tuxtla Gutiérrez, Chiapas, in Mexico is an area prone to flooding caused by extreme precipitation events. The basin has little information in which the records (with the same time resolution) of precipitation and hydrometry match. The basin model representing an area of 355.07 km$^2$ was characterized in the Stormwater Management Model (SWMM). The sensitivity analysis was performed for eight hydrological parameters and one hydraulic for two precipitation events and their impact on the depths of the Sabinal River. Based on the analysis, the parameters derived from the analysis that stand out as sensitive are the Manning coefficient of impervious surface and the minimum infiltration speed with $R^2 > 0.60$. The results obtained demonstrate the importance of knowing the sensitivity of the parameters and their selection to perform an adequate calibration.

**Keywords:** sensitivity analysis; rainfall–runoff model; parameters model

## 1. Introduction

Rainfall–runoff phenomena are one of the main processes within the hydrological cycle. In urban zones, the increases in imperviousness cause increased runoff, originating floods. Therefore, to protect the population and movable and immovable property, hydraulic structures that make up urban drainage (storm hydrants, collectors, emitters, and retention works, among others) are analyzed and designed. According to Jha et al. [1], the number of reported flood events affected people and associated economic damage has been significantly increasing over the past two decades. One tool in the analysis and design of these structures is the use of the urban drainage models, which have been developed for the last 30 years to contribute to the management and planning of stormwater [2,3]. The SWMM (Storm Water Management Model) was developed by the Environmental Protection Agency [4] to simulate the rainfall–runoff process in urban watersheds and is widely used for urban planning, analysis and design related to drainage systems [5–7]. Numerous studies have investigated the use of this model

to describe different phenomena related to runoff in urban basins; for example, Randall et al. [8] implemented SWMM to assess the behavior of runoff under Low Impact Development (LID) scenarios. Chang et al. [9] evaluated the DENFIS (Dynamic Evolving Neural Fuzzy Inference System) model performance compared with the physically based model SWMM. Agarwal and Kumar [10] implemented a runoff model to determine flood impact using the Green-Ampt Infiltration model in SWMM.

Technological advances in hydrological modeling have incorporated the use of data with greater detail in terms of spatial and temporal resolution. On the other hand, remote measurement techniques allow information to be obtained quickly in large areas through sensors operating in different spectral bands, which opens the door to the use of large amounts of data applied to hydrological models, which thereby become robust [11]. The ability to incorporate spatially distributed digital information is then hampered by a lack of data on the same time scale (precipitation measurement and runoff) [12,13]. However, modeling hydrological processes can be challenging, particularly in highly heterogeneous urbanized areas (land-use variation, slope, coverage) that produce multiple interactions between urban drainage structures and system (for wastewater and stormwater) at different temporal and spatial scales, which increases data requirements and complexity [14]. These complexities, in addition to the data shortage at the required level, make it difficult to define a universal methodology for reproducing urban flows at the catchment scale.

Hydrological models are approximations of natural systems, which create a substantial discrepancy between the results of the model and reality [15]. The results of the models need to be adjusted by means of parameter calibration, which helps to match the predictions with the corresponding observations [16–18]. The increase in the number of parameters that are adjusted in a model leads to a greater workload in the calibration process [19,20]. Therefore, to increase the speed of the process, it is important to perform a sensitivity analysis to learn the set of parameters to which the models are sensitive, to understand their behavior against their values' variation and to use this information to limit the number of parameters in the calibration [21,22]. It is therefore recommended to perform a sensitivity analysis before starting with hydrological modeling [23,24]. This analysis has been applied in different levels of watersheds. Shin and Choi [25] found that the size of the catchment makes a difference in the parameter sensitivity between rainfall events.

Some researchers, such as Mannina and Viviani [26], considered sensitivity analysis, identifying the model's most sensitive parameters. They applied the analysis to 17 parameters that influence the results of the discharges and concentrations of urban drainage, managing to reduce the number of parameters to 12, which were subsequently used for the calibration of the model. Kleidorfer et al. [27] analyzed the impact of uncertainty on modeling input data considering the parameters with greater sensitivity, using the Metropolis–Hastings algorithm for the assessment of sensitivity in the calibration parameters. Bárdossy [28] mentions that hydrological parameters cannot be identified as a single set of values, and changes to a parameter can be absorbed by the remaining parameters of the model. Other researchers, such as Thorndahl et al. [29], performed a sensitivity analysis of a set of parameters, by comparing the conditions of the conceptual model and the general model, finding that the parameter of greater sensitivity is the hydrological reduction factor. Bajracharya et al. [30] performed a global sensitivity analysis (using the Variogram Analysis of Response Surfaces technique) of the parameters that govern the behavior of runoffs of the Nelson Churchill River basin, represented in the Hydrological Predictions for the Environment model (HYPE). Other studies that perform sensitivity analysis applied to the SWMM model reveal the behavior of the parameters according to the study area and its characterization [31–34].

Due to data being scarce in most of the world (especially in developing countries), research should focus on the reliability of hydrological models. This can be achieved by comparing different sensitivity analysis approaches in data-scarce regions.

This research presents a methodology that uses four common expressions to assess the sensitivity of hydrological model parameters in an urban basin in Mexico. The expressions used are absolute–relative sensitivity, relative–absolute sensitivity, relative–relative sensitivity, and correlation coefficient $R^2$.

This use of expressions aims to show their performance together to determine the sensitivity of parameters in a well-known open-source model (SWMM). This application will show the benefits of implementing such an analysis in applications with limited data. This work builds on our previous study [35], in which a storm with a single peak was evaluated as input data and a sensitivity index was calculated. In this work, a more robust sensitivity analysis is performed, as parameter influence is evaluated with more than one equation. In addition, two different storms were evaluated, allowing us to observe the difference in behavior of the riverbed depths due to the parameters under these differing conditions. The study comprises the analysis of nine parameters that are used for modeling and are difficult to estimate because of their complexity and variability in surface coverage in an urban context.

The document is organized as follows: Section 2 illustrates the materials and methods used in sensitivity analysis; Section 3 presents the results of the analysis; Section 4 presents the discussion on the results. Finally, Section 5 presents the main conclusions arising from this work.

## 2. Materials and Methods

### 2.1. Study Area

The Sabinal River basin has 355.07 km$^2$ of surface, is located between the coordinates 16°42′ and 16°54′ north and between the latitude 93°20′ and 9°02′ west, with an elevation in the range of 384 to 1064 m above sea level and an average elevation of 724 m above sea level. This basin is characterized by 42.31% permeable soil and 57.69% waterproof soil, the latter representing urban areas. In general, the basin has an average slope of 6.89% and a concentration time of 328.80 min. The city of Tuxtla Gutiérrez, Chiapas in México is located within the Sabinal River basin and is crossed from west to east by the main riverbed, 21 km long. It has flooding problems caused by extreme precipitation events, which can occur mainly from May to October. Therefore, sensitivity analysis was carried out using the extreme precipitation events of 07/10/2011 (Event 1) and 07/27/2011 (Event 2), recorded every 10 min by seven automatic stations. Figure 1 shows the automatic stations (yellow circles), as well as the urban basin of the Sabinal River, which was divided into 96 sub-basins for modeling. Tables 1 and 2 show the properties of the precipitation events mentioned above, in view of Figure 2a,b.

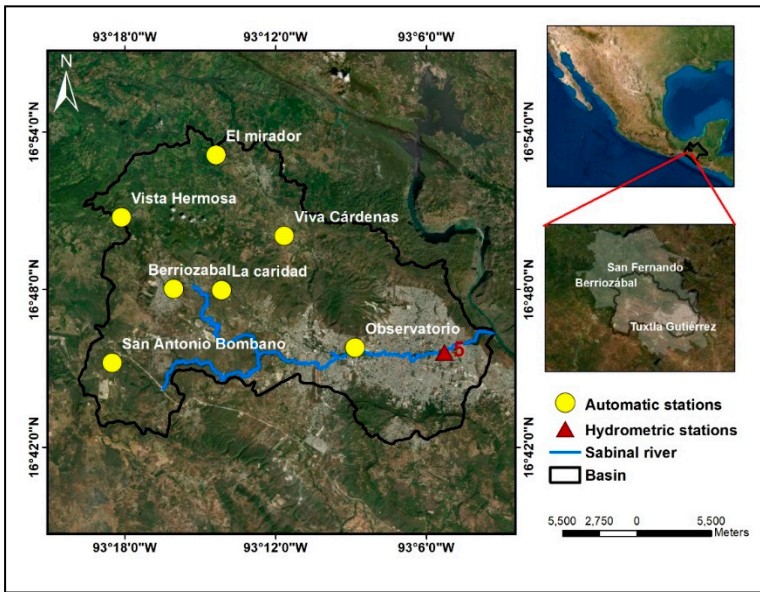

**Figure 1.** Urban basin of Tuxtla Gutiérrez, Chiapas, México.

**Table 1.** Precipitation characteristics, Event 1.

| Station | Duration (min) | Δt (min) | Maximum Intensity, (mm/h) | Accumulated Precipitation (mm) |
|---|---|---|---|---|
| Berriozábal | 490.00 | 10.00 | 100.50 | 45.50 |
| Caridad | 490.00 | 10.00 | 117.00 | 53.75 |
| Mirador | 490.00 | 10.00 | 81.00 | 27.00 |
| Observatorio | 490.00 | 10.00 | 115.50 | 23.75 |
| San Antonio Bombanó | 490.00 | 10.00 | 78.00 | 32.25 |
| Vista Hermosa | 490.00 | 10.00 | 36.00 | 33.25 |
| Viva Cárdenas | 490.00 | 10.00 | 79.50 | 35.75 |

**Table 2.** Precipitation characteristics, Event 2.

| Station | Duration (min) | Δt, (min) | Maximum Intensity, (mm/h) | Accumulated Precipitation (mm) |
|---|---|---|---|---|
| Berriozábal | 1820.00 | 10.00 | 57.00 | 22.50 |
| Caridad | 1820.00 | 10.00 | 70.50 | 43.75 |
| Mirador | 1820.00 | 10.00 | 84.00 | 44.25 |
| Observatorio | 1820.00 | 10.00 | 81.00 | 58.75 |
| San Antonio Bombanó | 1820.00 | 10.00 | 19.50 | 16.50 |
| Vista Hermosa | 1820.00 | 10.00 | 51.00 | 32.00 |
| Viva Cárdenas | 1820.00 | 10.00 | 40.50 | 22.25 |

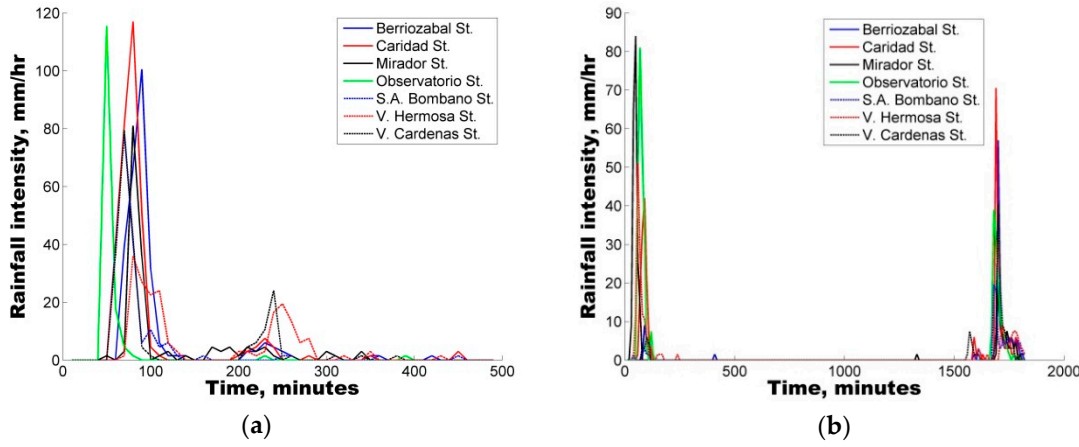

**Figure 2.** (**a**) Precipitation Event 1 and (**b**) Precipitation Event 2.

Like precipitation data, data from the Parque del Oriente hydrometric station are available at the exit of the basin, which has depth information for the month of July with registration every 10 min in the river section. Figure 1 shows the location of the hydrometric station with the number 5 and with the triangle symbol in red. Figure 3a,b show the depths recorded by that station caused by precipitation events 1 and 2.

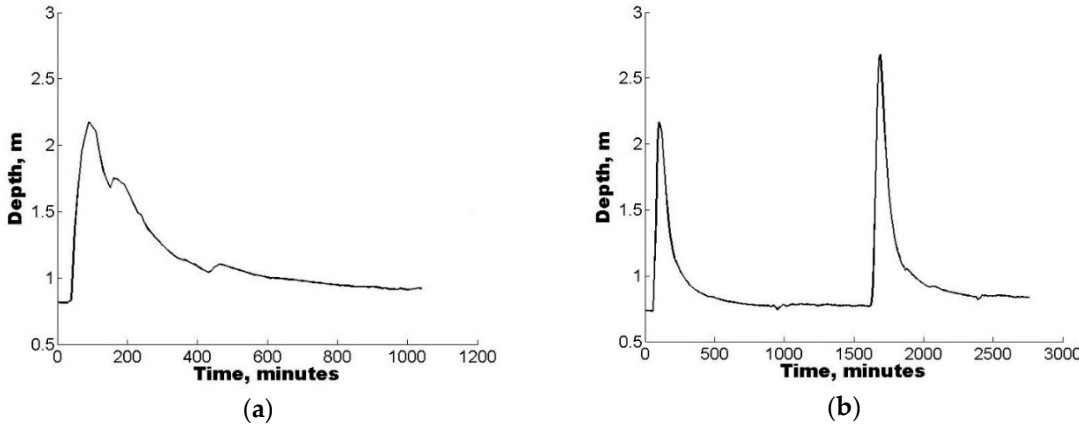

**Figure 3.** Sabinal river hydrometry in Parque del Oriente station; (**a**) Event 1 and (**b**) Event 2.

### 2.2. Hydrological Model

The integrated Stormwater Management Model (SWMM) was generated by the Environmental Protection Agency in the United States [5]. SWMM is a dynamic hydrology–hydraulic water quality simulation model that can be used for single-event or long-term (continuous) simulation of runoff quantity and quality from primarily urban areas [4,5,8]. The runoff component operates on a collection of sub-catchment areas that receive precipitation and generate runoff. Each one of the sub-basins is considered as a nonlinear reservoir, where the contribution of flow to the deposits comes from precipitation, snow or releases from upstream stores. SWMM also simulates route flows from the system, infiltration, evaporation and surface runoff. The surface runoff of a given area is determined when the water depth within a catchment exceeds the maximum storage value, and the outflow is determined by Manning (1), which integrates the continuity Equation (2), considering friction through the incorporation of the Manning coefficient (n). The major loss considered in the rainfall and runoff modeling is infiltration loss. In this study, infiltration loss is calculated with the Horton Equation (3) [36]. The infiltration losses are considered only from the pervious areas of a sub-catchment [5].

$$Q_M = \frac{1}{n} L \, (y - y')^{5/3} s^{1/2} \tag{1}$$

Here, $Q_M$ is the flow by Manning, $n$ is the Manning coefficient, $L$ is the width sub-basin, $y$ is the water depth, $y'$ is the lowering of height storage and $s$ is the slope.

$$A \frac{\partial y}{\partial t} = Ai - Q \tag{2}$$

Here, $Q$ is the flow, $A$ is the area of the basin, $i$ is the intensity of the rain, $y$ is the depth of storage in depressions and $t$ is the time.

$$f_p = f_0 - (f_0 - f_\infty)e^{-\propto_d(t - t_w)} \tag{3}$$

Here, $f_p$ is the soil infiltration capacity, $f_\infty$ is the minimum or end value of $f_p$ ($in \, t = \infty$), $f_0$ is the maximum or initial value of $f_p$ ($in \, t = 0$), $t_w$ is the start time of the storm and $\propto_d$ is the decay coefficient.

SWMM requires the input of parameters related to catchment characteristics, sewer network and soil type. The values range of the parameters was derived from the following (Table 3, [37,38]): Manning's roughness for overland surfaces and conduits, soil infiltration parameters and surface depression storage. Manning's roughness is the measure of resistance to the runoff flow. The value of the roughness coefficient depends on the type of soil, surface cover and vegetation in pervious areas, and in impervious areas it depends on the type of the material used in the construction of streets and building roofs. Other parameters depend on the soil type and the slope; for example, the impervious area depression storage, which is defined as water stored in depressions on impervious areas (depleted

only by evaporation), and pervious area depression storage is defined as water stored in depressions on pervious areas (subject to infiltration and evaporation). Likewise, the measure of urbanization is given as the percentage of imperviousness for each sub-catchment or area depression storage, where the urban catchments are composed of pervious and impervious areas (which increases when the urban area develops). For Horton's infiltration equation, the values of minimum or maximum infiltration rate and decay coefficient depend on the soil, vegetation and initial moisture content; these parameters should be estimated using results from the field or from specialized literature. Finally, Manning's roughness coefficient for conduits is one of the parameters used to calculate flow in a pipe or open channel and depends on the material type.

**Table 3.** SWWM model parameters.

| Parameter | Abbreviation | Maximum Value | Minimum Value |
|---|---|---|---|
| Manning's duct coefficient | ManN | 0.010 | 0.030 |
| Manning's N for impervious area | Nimperv | 0.001 | 0.200 |
| Manning's N for pervious area | Nperv | 0.010 | 0.200 |
| Depth of depression storage on impervious area, mm | Simperv | 0.000 | 10.000 |
| Depth of depression storage on pervious area, mm | Sperv | 0.000 | 20.000 |
| Percentage of impervious area with no depression storage, % | PctZero | 0.000 | 100.000 |
| Maximum infiltration rate, mm/hr | MaxRate_fa | 1.000 | 200.000 |
| Minimum infiltration rate, mm/hr | MinRate_fe | 1.000 | 25.000 |
| Decay coefficient, 1/hr | Decay_k | 1.000 | 30.000 |

### 2.3. Sensitivity Analysis

Parameter sensitivity analysis is a necessary background for any deeper analysis and helps to improve the understanding of the model's behavior. Its goal is to explore the change in model output resulting from a change in model parameters or model inputs and to separate influential from non-influential parameters. Sensitivity analysis investigates the sensitivity of a parameter with respect to the simulation results at a certain parameter value. The following expressions calculate different sensitivity indexes for each of the possible model parameters [39,40]:

$$s_{i,j}(\Theta_M) = \Theta_{M,j} \frac{\partial f(\Theta_{M,j})}{\partial \Theta_{M,j}} \tag{4}$$

$$s_{i,j}(\Theta_M) = \frac{1}{f(\Theta_{M,j})} \frac{\partial f(\Theta_{M,j})}{\partial \Theta_{M,j}} \tag{5}$$

$$s_{i,j}(\Theta_M) = \frac{\Theta_{M,j}}{f(\Theta_{M,j})} \frac{\partial f(\Theta_{M,j})}{\partial \Theta_{M,j}} \tag{6}$$

where $f(\Theta_{M,j})$ represents the n output variables of the model, and $\Theta_{M,j}$ represents the *j*th independent parameters of the model.

Expression (4) represents the absolute–relative sensitivity, which describes the absolute change in the results for a relative change in parameters. Expression (5) is the relative–absolute sensitivity, which describes the relative change in the results for an absolute change of the parameter. Finally, expression (6) is the relative–relative sensitivity, which describes the relative change in results for a relative change in parameters [39,40].

The gradient term $\partial f(\Theta_{M,j})/\partial \Theta_{M,j}$ is solved numerically by using runs of the model with slightly different values of $\Theta_M$. Then the gradient term can be approximated by expression (7):

$$\frac{\partial f(\Theta_{M,j})}{\partial \Theta_{M,j}} = \frac{f(\Theta_{M,j} + \Delta\Theta_{M,j}) - f(\Theta_{M,j} - \Delta\Theta_{M,j})}{2\Delta\Theta_{M,j}} \tag{7}$$

where $\Delta\Theta_{M,j}$ is a small increment in the parameter value.

In addition to the three equations above, the coefficient of determination $R^2$ calculation is used. This is defined as the squared value of the coefficient of correlation [41].

$$R^2 = \left( \frac{\sum_{i=1}^{n}(O_i - \overline{O})(P_i - \overline{P})}{\sqrt{\sum_{i=1}^{n}(O_i - \overline{O})^2} \sqrt{\sum_{i=1}^{n}(P_i - \overline{P})^2}} \right)^2 \tag{8}$$

Here, $O$ are the observed values and $P$ are the modeling values.

The range of $R^2$ lies between 0 and 1 and describes how much of the observed dispersion is explained by the prediction. A value of 0 means no correlation, whereas a value of 1 means that the dispersion of the prediction is equal to that of the observation [41].

### 2.4. Methodology

The general methodology set out in this work for sensitivity analysis is shown in Figure 4. The methodology indicates that the sensitivity analysis process begins with the characterization of the urban basin in which the runoffs (depths in the riverbed) promoted by the precipitation event are evaluated. Topographic, hydrographic and land-use characteristics were configured.

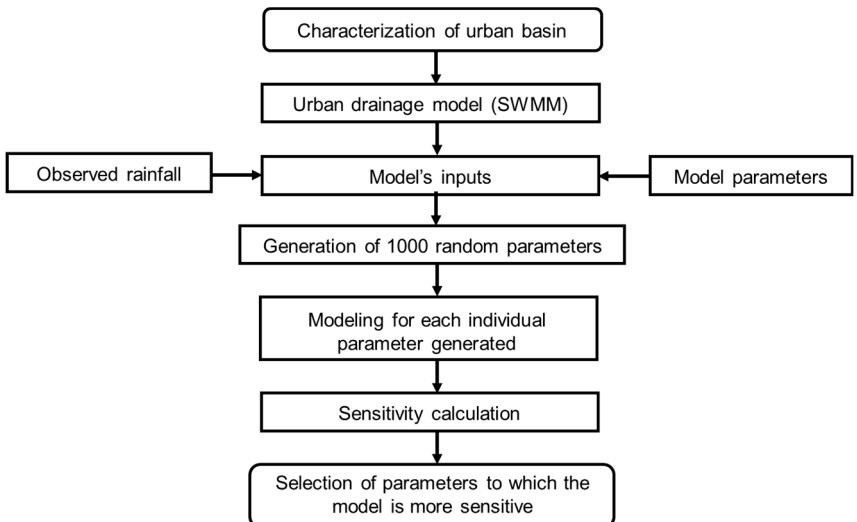

**Figure 4.** Sensitivity analysis methodology.

The hydrological model SWMM is then selected to perform runoff modeling. The input data required for modeling are then entered—in this case, precipitation events and parameters.

The next step is the individual sensitivity analysis of the input, hydrological and hydraulic parameters. The analysis is performed by evaluating the runoff outputs of the model, generated by 1000 individual values of one of the modeling parameters (randomly generated with the Monte Carlo Simple method), and keeping the values of the other parameters fixed. The process is repeated for each of the parameters. In the next step, the sensitivity associated with the modeling results generated by the variation of individual parameters is calculated with Equations (4)–(6).

Obtaining sensitivity of the parameters includes doing the following:

- Obtaining box plot from the results of accumulated depths and calculation of mean, standard deviation and coefficient of variation in order to evaluate the dispersion of the data.
- Obtaining box plot of parameters and calculation of mean, standard deviation and variance in order to evaluate data dispersion.

Once the sensitivity is calculated, $R^2$ is obtained in order to evaluate the performance of the results due to the parameters (Equation (8)).

Finally, validation of sensitivity analysis results is carried out using event 2 and repeating the steps of this methodology.

## 3. Results

### 3.1. Sensitivity Analysis Event 1

This section shows the results of the sensitivity analysis carried out in the urban basin of the Sabinal River using the Equations (4)–(6) and (8) described above. The model's simulations were carried out using input data (hydrological and hydraulic parameters and precipitation Event 1) in SWMM 5.0. In addition, hydrometric station 5 is represented in the model, from which the results of the cumulative depth series corresponding to the variation of the parameters (1000 embodiments per parameter) were taken.

Figure 5 shows the variation of the accumulated depths with respect to the parameters. The accumulated output depth values for MinRate_fe are in a range of 167.03 to 218.56 m, and Nimperv from 169.94 to 243.66 m, approximately. The accumulated depth values, generated by the MinRate_fe parameter, within quartile two (179.03) are the closest to the accumulated depth (183.33 m) of the hydrometric series observed in the Sabinal. The Nimperv parameters that generate depth values closer to those observed (Figure 5) are observed between the lower limit and quartile one (169.94 to 210.25). On the other hand, the parameters Decay_k, MaxRate_fa, PctZero, Sperv, Simperv and Nperv generate output results with less variation between them. As for the variation of the ManN parameter, it does not generate a change in the modeling results.

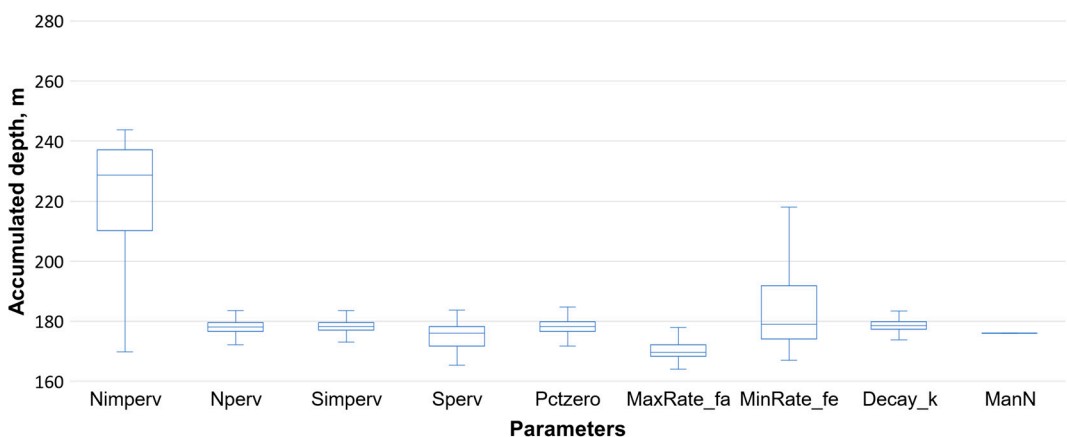

**Figure 5.** Variation of the accumulated depth on the Sabinal River with respect to the parameters.

A key feature of Figure 5 is that MinRate_fe has a positive bias of the accumulated depths, since the case's whisker is longer towards the high values (there is greater dispersion of quartile three at the upper limit of the data), while Nimperv has a negative bias. This is because the box whiskers are long towards the low values (greater dispersion of the lower limit to the quartile one data).

In Table 4, it can be noted that the results obtained from accumulated depths have greater variation with respect to the MinRate_fe and Nimperv parameters since the standard deviation and coefficient of variation are 20.09 m and 10.73%, respectively, for MinRate_fe, and for Nimperv, the standard deviation is 19.85 m and coefficient of variation is 8.97%.

**Table 4.** Mean μ, standard deviation σ, and coefficient of variation Cv of the depth values with respect to the parameters, Event 1.

| Parameter | μ (m) | σ (m) | Cv (%) |
|---|---|---|---|
| Nimperv | **221.30** | **19.85** | **8.97** |
| Nperv | 178.05 | 2.15 | 1.21 |
| Simperv | 178.31 | 1.96 | 1.10 |
| Sperv | 175.10 | 3.89 | 2.22 |
| PctZero | 177.75 | 3.23 | 1.82 |
| MaxRate_fa | 170.61 | 3.34 | 1.96 |
| MinRate_fe | **187.25** | **20.09** | **10.73** |
| Decay_k | 178.57 | 1.85 | 1.04 |
| ManN | 176.08 | 0.00 | 0.00 |

Once the analysis of the accumulated depths had been done, sensitivity analysis was carried out for each of the expressions presented. Figure 6 shows that the rate of change of the random parameters used in this analysis generated different values of absolute–relative sensitivity indexes, except ManN, the value of which was zero because it had no variation in accumulated depth. As a result, in quartile three at the upper limit, the parameters MinRate_fe and Nimperv have greater dispersion of the high values (positive bias) and of absolute-relative sensitivity values: 0.0810 to 0.1890 and 0.0761 to 0.1767, respectively.

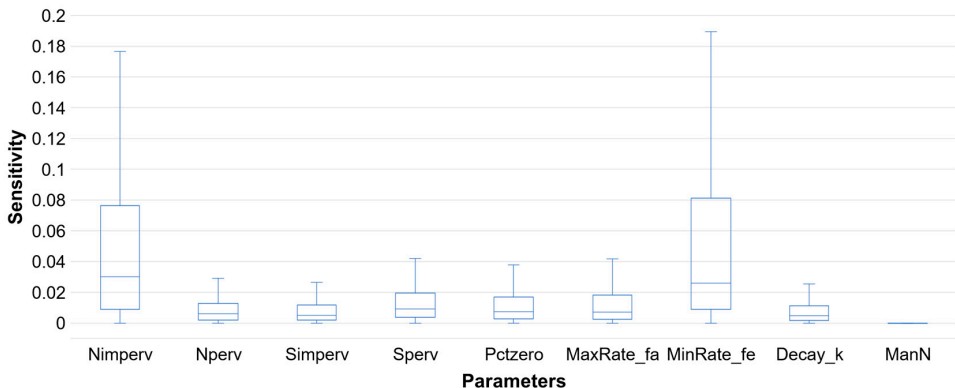

**Figure 6.** Comparison of parameters with respect to the absolute–relative sensitivity index, Event 1.

On the other hand, Figure 7 shows the results of the relative–absolute sensitivity. The Nimperv and Nperv parameters have high dispersion values: Nimperv from 1.1232 to 2.1777 (quartile three at upper limit) and Nperv from 0.2231 to 0.4943 (quartile three to upper limit), both parameters with positive bias. All other parameters have low sensitivity values with less dispersion.

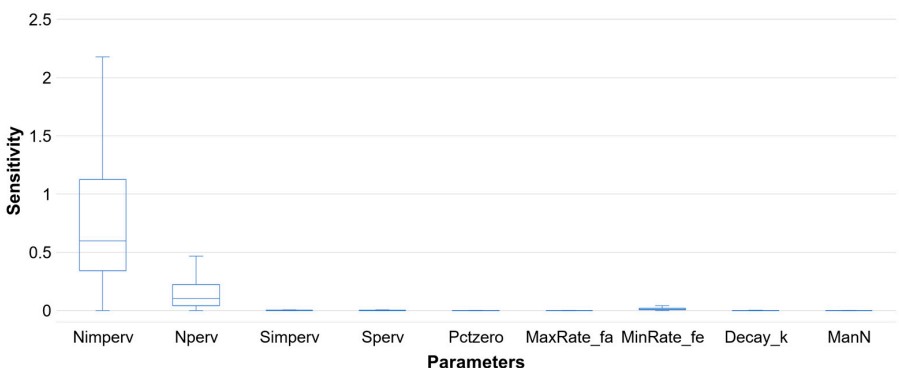

**Figure 7.** Comparison of parameters with respect to the relative–absolute sensitivity index, Event 1.

Finally, Figure 8 shows the sensitivity values of the parameters obtained from the relative–relative relationship: the Nimperv and MinRate_fe parameters are those with high sensitivity values and with the highest dispersion ranging from quartile two to the upper limit. The sensitivity values for Nimperv range from 0.0482 to 0.1635 and for MinRate_fe from 0.0554 to 0.2180. As in calculations with previous sensitivity calculation expressions, the other parameters have less bias in the obtained values.

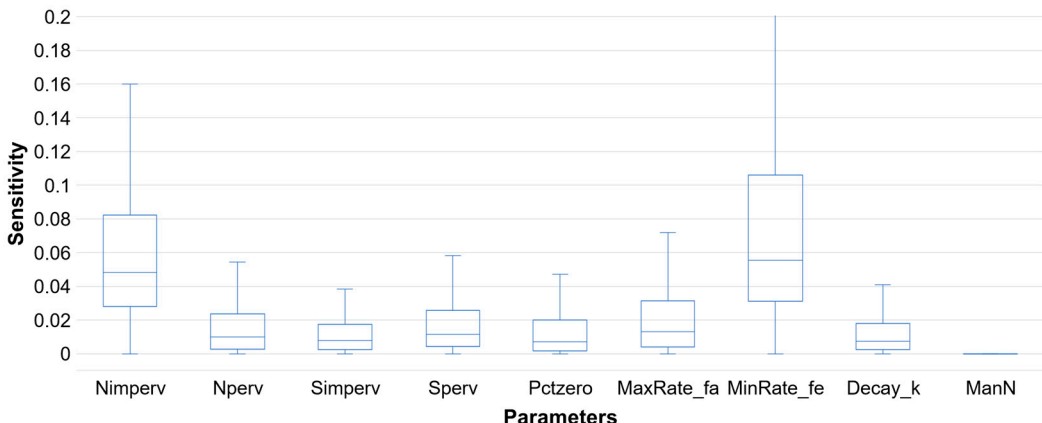

**Figure 8.** Comparison of parameters with respect to the relative–relative sensitivity index, Event 1.

Table 5 shows the statistical characteristics of each of the above figures, which correspond to the different methods to obtain the sensitivity of the model results according to the parameters. For example, the absolute–relative sensitivity values of the Nimperv parameter are 0.1047 and 0.0110 of standard deviation and variance, respectively; 0.5723 for standard deviation and 0.3276 for variance in the case of relative–absolute sensitivity; and for relative–relative sensitivity, 0.04870 for standard deviation and 0.00237 for variance. In the case of MinRate_fe for the relative–absolute sensitivity, the values are 0.2223 and 0.00494 for standard deviation and variance, respectively, and for the relative–relative sensitivity they are 0.08626 for standard deviation and 0.00744 for variance. In the case of relative–absolute sensitivity, instead of MinRate_fe, it was the Nperv parameter with values of 0.3961 for standard deviation and 0.1569 for variance.

**Table 5.** Mean $\mu$, standard deviation $\sigma$ and variance, $\sigma^2$ of sensitivity values, Event 1.

| Parameter | Absolute–Relative | | | Relative–Absolute | | | Relative–Relative | | |
|---|---|---|---|---|---|---|---|---|---|
| | $\mu$ | $\sigma$ | $\sigma^2$ | $\mu$ | $\sigma$ | $\sigma^2$ | $\mu$ | $\sigma$ | $\sigma^2$ |
| Nimperv | 0.0670 | **0.1047** | **0.0110** | 0.7875 | **0.5723** | **0.3276** | 0.05945 | **0.04870** | **0.00237** |
| Nperv | 0.0139 | 0.0491 | 0.0024 | 0.2332 | 0.3961 | 0.1569 | 0.02644 | 0.06645 | 0.00442 |
| Simperv | 0.0116 | 0.0241 | 0.0006 | 0.0061 | 0.0262 | 0.0007 | 0.02112 | 0.05493 | 0.00302 |
| Sperv | 0.0184 | 0.0424 | 0.0018 | 0.0041 | 0.0192 | 0.0004 | 0.02677 | 0.05586 | 0.00312 |
| Pctzero | 0.0187 | 0.0419 | 0.0018 | 0.0009 | 0.0066 | 0.0000 | 0.02659 | 0.07668 | 0.00588 |
| Maxrate_fa | 0.0207 | 0.0695 | 0.0048 | 0.0007 | 0.0042 | 0.0000 | 0.03378 | 0.06498 | 0.00422 |
| Minrate_fe | 0.0970 | **0.2223** | **0.0494** | 0.0202 | **0.0660** | **0.0044** | 0.08293 | **0.08626** | **0.00744** |
| Decay_k | 0.0109 | 0.0240 | 0.0006 | 0.0026 | 0.0106 | 0.0001 | 0.02447 | 0.07148 | 0.00511 |
| ManN | 0.0000 | 0.0000 | 0.0000 | 0.0000 | 0.0000 | 0.0000 | 0.00000 | 0.00000 | 0.00000 |

Based on the above analysis, Figure 9a (medium sensitivity) and Figure 9b (maximum sensitivity, according to the upper limit of the whisker box) show the most sensitive parameters in the hydrological simulation model.

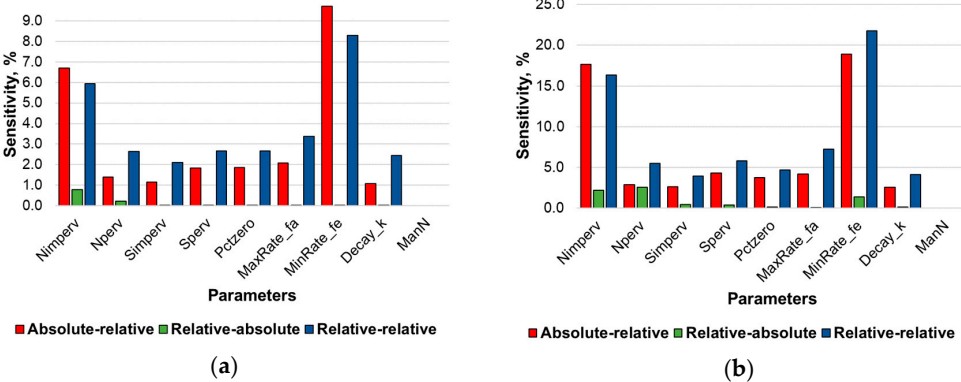

**Figure 9.** (**a**) Average sensitivity of modeling parameters and (**b**) maximum sensitivity of modeling parameters, Event 1.

$R^2$ was calculated according to the methodology. In Figure 10, $R^2$ efficiency results are shown: it can be seen that the $R^2$ values for Nimperv are in the range of 0.31 to 0.82, having greater amplitude in the whiskers box. The next parameter with greater amplitude in the whiskers box is MinRate_fe, with a range of $R^2$ from 0.62 to 0.79. The other parameters have less bias in the obtained values.

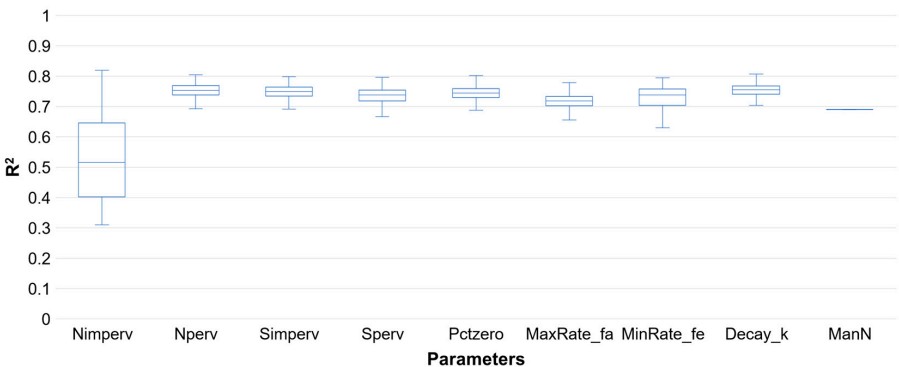

**Figure 10.** Comparison of parameters with respect to $R^2$, Event 1.

Based on the sensitivity and $R^2$ calculation, Figure 11a,b show the behavior of the depths generated by Event 1. This shows the change of the depth hydrogram according to the maximum, minimum and mean value and the parameter value with greater $R^2$, for parameters with higher Nimperv sensitivity and MinRate_fe.

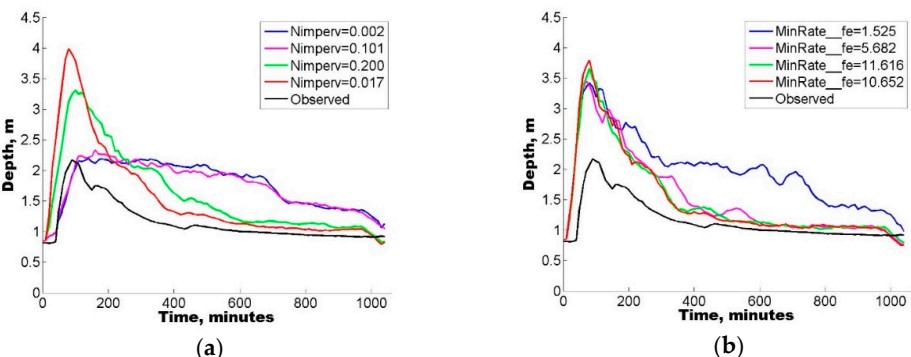

**Figure 11.** Depth levels of Sabinal river using (**a**) Nimperv parameter and (**b**) MinRate_fe parameter. The blue line is a minimum value, the magenta line is an average value, the green line is a maximum value, the red line is a parameter with a maximum value of $R^2$ and the black line is depth levels in hydrometric station.

In contrast to Figure 12a,b, the change in the depth hydrogram is observed according to the maximum, minimum, mean value and the parameter value with the highest $R^2$ for two of the parameters with lower Simperv sensitivity and Decay_k.

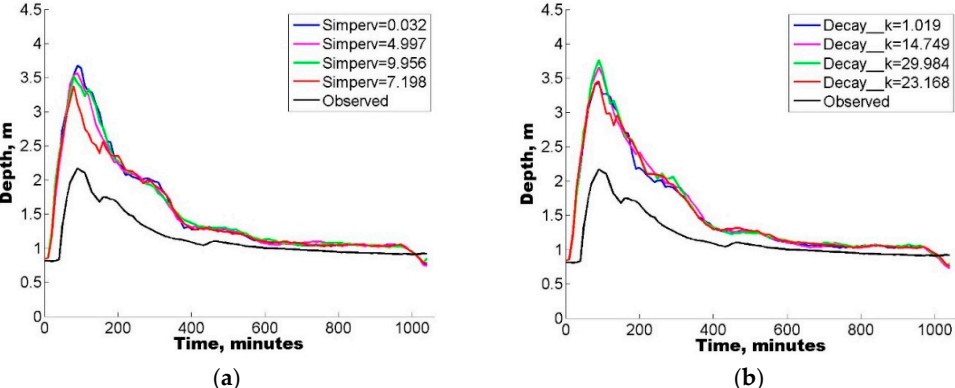

(a)          (b)

**Figure 12.** Depth levels of Sabinal river using (**a**) Simperv parameter and (**b**) Decay_k parameter. The blue line is a minimum value, the magenta line is an average value, the green line is a maximum value, the red line is a parameter with maximum value of $R^2$ and the black line is depth levels in hydrometric station.

### 3.2. Validation of Results, Event 2

This section shows the results of the sensitivity analysis carried out in the urban basin of the Sabinal River according to the methodology proposed (expressions (4), (5), (6) and (8)). The modeling was carried out with the input data (hydrological and hydraulic parameters and precipitation Event 2). In this case, the accumulated depth of the observed hydrometry was 201.25 m.

Figure 13 shows the variation of the accumulated depths with respect to the parameters, and it can be observed that the parameter that generates the greatest variation in the accumulated output depth values is Nimperv, which is in a range of 371.58 to 496.41 m. The accumulated depth values, generated by the Nimperv parameter, within the lower bound and quartile two (371.58 to 437.20), are the most dispersed and close to the observed accumulated depth. The MinRate_fe parameter generated depth values close to observed and less dispersion than the previous parameter (Figure 13) are observed between the lower bound and the upper bound (367.84 to 384.52). On the other hand, the parameters MaxRate_fa, PctZero, Decay_k, Sperv, Simperv and Nperv generate output results with less variation between them (compact boxes). As for the variation of the ManN parameter, these do not result in a change in modeling results.

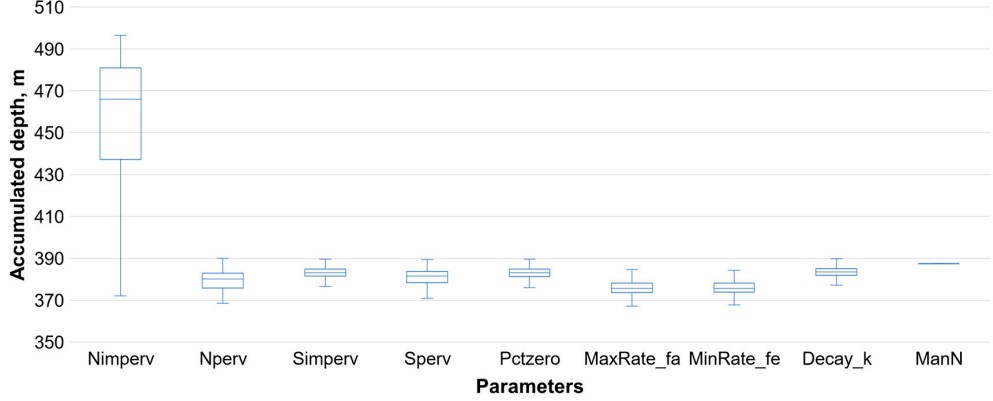

**Figure 13.** Variation of the accumulated depth on the Sabinal River with respect to the parameters.

One key feature in Figure 13 is that for Nimperv, it has a negative bias of the accumulated depths, since the whisker of the box is longer towards the low values (there is greater dispersion of the lower limit to quartile one of the data). With respect to the above, in Table 6, it can be noted that the results obtained from accumulated depths have greater variation with respect to the Nimperv parameter, since the standard deviation and coefficient of variation are 32.47 m and 7.12%. For MinRate_fe, the standard deviation is 4.89 m and the coefficient of variation is 1.30%.

**Table 6.** Mean μ, standard deviation σ, and coefficient of variation Cv of the depth values with respect to the parameters, Event 2.

| Parameter | μ (m) | σ (m) | Cv (%) |
|---|---|---|---|
| Nimperv | **455.74** | **32.47** | **7.12** |
| Nperv | 379.46 | 4.38 | 1.15 |
| Simperv | 383.17 | 2.60 | 0.68 |
| Sperv | 380.89 | 3.85 | 1.01 |
| PctZero | 383.11 | 2.66 | 0.69 |
| MaxRate_fa | 376.21 | 3.44 | 0.91 |
| MinRate_fe | 376.43 | 4.89 | 1.30 |
| Decay_k | 383.45 | 2.38 | 0.62 |
| ManN | 387.46 | 0.00 | 0.00 |

On the other hand, the sensitivity results indicate that parameter Nimperv has greater dispersion (positive bias) in quartile three and the upper limit (Figure 14), where the values of absolute–relative sensitivity range from 0.0654 to 0.1514, respectively. In contrast, the parameters MinRate_fe, Decay_k, MaxRate_fa, PctZero, Sperv, Simperv and Nperv have sensitivity values with less dispersion.

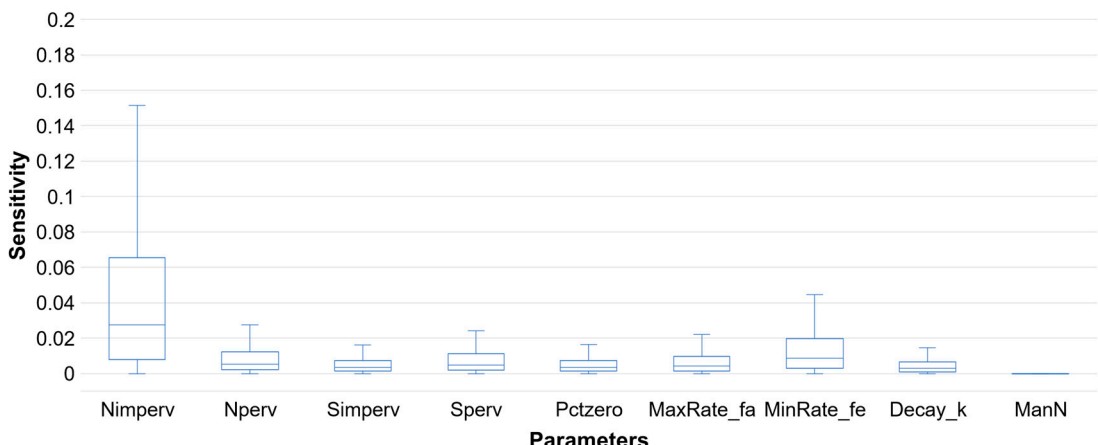

**Figure 14.** Comparison of parameters with respect to the absolute–relative sensitivity index, Event 2.

Figure 15 shows the results of the relative–absolute sensitivity: the Nimperv and Nperv parameters are the ones with the highest sensitivity values with respect to the model outputs; in this case, Nimperv from 0.7863 to 1.4513 (quartile three at upper limit) and Nperv from 0.2231 to 0.4943 (quartile three to upper limit), both parameters with positive bias. All other parameters have low sensitivity values with less dispersion.

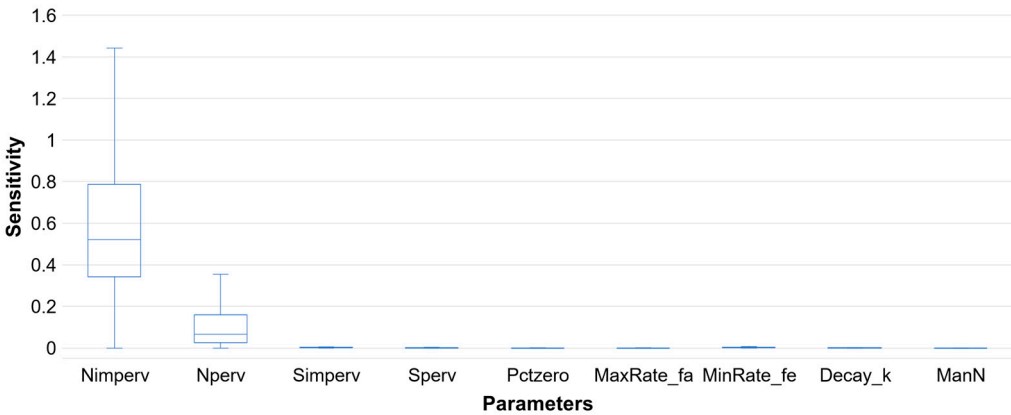

**Figure 15.** Comparison of parameters with respect to the relative–absolute sensitivity index, Event 2.

Finally, Figure 16 shows the sensitivity values of the parameters obtained from the relative–relative relationship: the Nimperv parameter is the one with high sensitivity values with the highest dispersion ranging from quartile three to the upper limit. In this case, the sensitivity values for Nimperv range from 0.0679 to 0.1326. As in calculations with previous sensitivity calculation expressions, the other parameters have less dispersion in the obtained values.

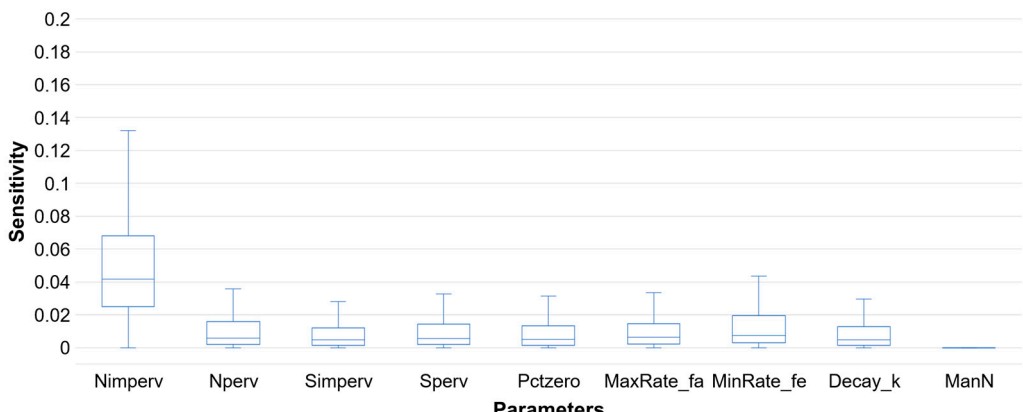

**Figure 16.** Comparison of parameters with respect to the relative–relative sensitivity index, Event 2.

Table 7 shows the statistical characteristics of each of the above figures, which correspond to the different methods of obtaining the sensitivity of the model results according to the parameters. For example, the Nimperv parameter for relative–absolute sensitivity has values of 0.1031 and 0.0106 of standard deviation and variance, respectively. On the other hand, these values are 0.6241 for the standard deviation and 0.3895 of variance with the relative–absolute sensitivity. Finally, for relative–relative sensitivity, these values are 0.05722 for standard deviation and 0.00327 for variance. In this case, the Nperv and Simperv parameters are not considered, because having a large amount of outlier data causes the standard deviation and sensitivity variance values to increase.

Based on the above analysis, Figure 17a,b show the medium sensitivity and maximum sensitivity (according to the upper limit of the whisker box) of parameters that generate the sensitivity in the hydrological simulation model.

**Table 7.** Mean, μ, standard deviation, σ, and variance, σ², of sensitivity values, Event 2.

| Parameter | Absolute–Relative | | | Relative–Absolute | | | Relative–Relative | | |
|---|---|---|---|---|---|---|---|---|---|
| | μ | σ | σ² | μ | σ | σ² | μ | σ | σ² |
| Nimperv | 0.0596 | **0.1031** | **0.0106** | 0.6875 | **0.6241** | **0.3895** | 0.05345 | **0.05722** | **0.00327** |
| Nperv | 0.0107 | 0.0221 | 0.0005 | 0.2182 | 0.6057 | 0.3668 | 0.02482 | 0.10692 | 0.01143 |
| Simperv | 0.0090 | 0.0445 | 0.0020 | 0.0059 | 0.0323 | 0.0010 | 0.03140 | 0.20057 | 0.04023 |
| Sperv | 0.0107 | 0.0294 | 0.0009 | 0.0023 | 0.0090 | 0.0001 | 0.02197 | 0.10039 | 0.01008 |
| Pctzero | 0.0090 | 0.0415 | 0.0017 | 0.0005 | 0.0028 | 0.0000 | 0.02181 | 0.07978 | 0.00637 |
| Maxrate_fa | 0.0108 | 0.0321 | 0.0010 | 0.0002 | 0.0008 | 0.0000 | 0.02257 | 0.10334 | 0.01068 |
| Minrate_fe | 0.0490 | 0.3743 | 0.1401 | 0.0057 | 0.0305 | 0.0009 | 0.03501 | 0.23318 | 0.05437 |
| Decay_k | 0.0086 | 0.0340 | 0.0012 | 0.0017 | 0.0108 | 0.0001 | 0.02140 | 0.10890 | 0.01190 |
| ManN | 0.0000 | 0.0000 | 0.0000 | 0.0000 | 0.0000 | 0.0000 | 0.00000 | 0.00000 | 0.00000 |

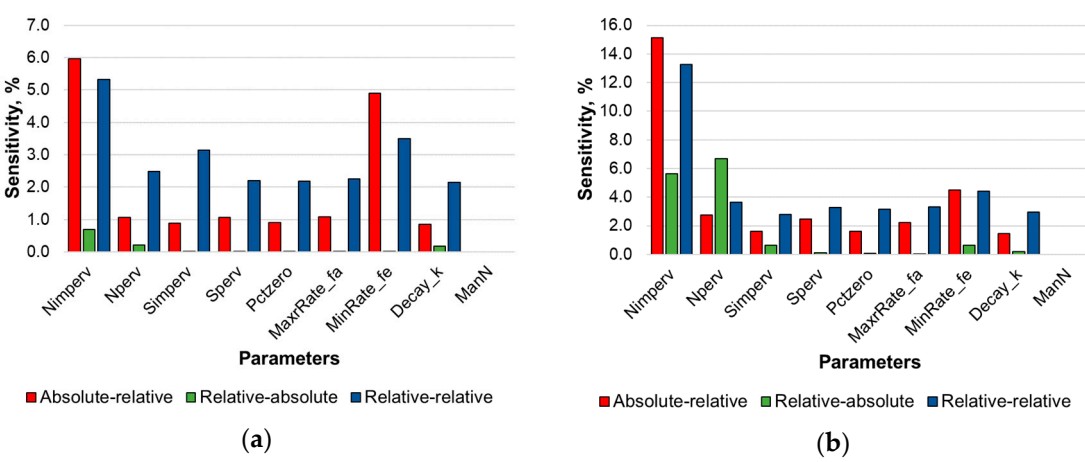

**Figure 17.** (**a**) Average sensitivity of modeling parameters and (**b**) maximum sensitivity of modeling parameters, Event 2.

$R^2$ was calculated according to the methodology. In Figure 18, $R^2$ efficiency results are shown, where the $R^2$ values for Nimperv are in the range of 0.35 to 0.74, having greater amplitude in the whiskers box. The other parameters have less bias in the obtained values.

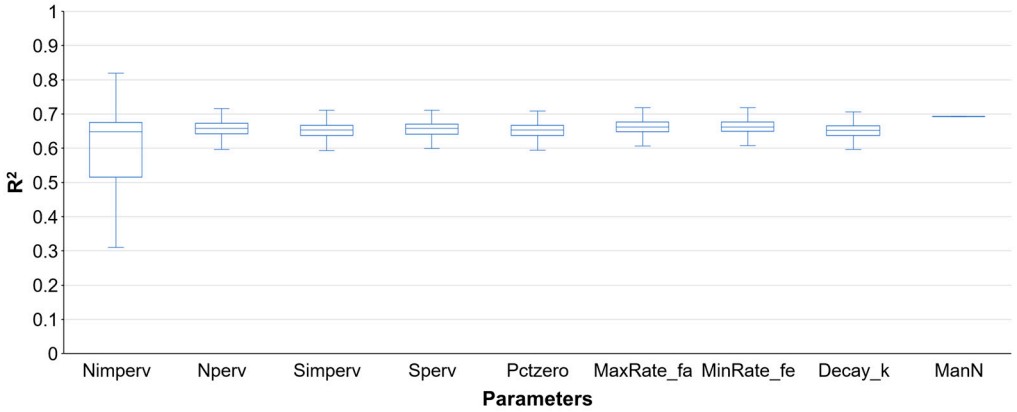

**Figure 18.** Comparison of parameters with respect to $R^2$, Event 2.

Based on the sensitivity and $R^2$ calculation, Figure 19a,b show the behavior of the depths generated by Event 1. This shows the change of the depth hydrogram according to the maximum, minimum, and mean values and the parameter value with greater $R^2$. These four parameters have higher Nimperv sensitivity and one of the lowest sensitivity parameters, MinRate_fe.

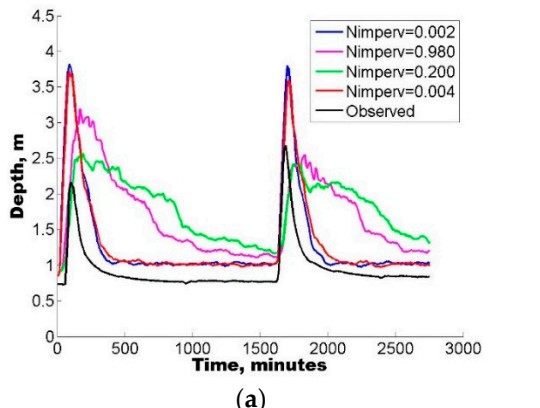
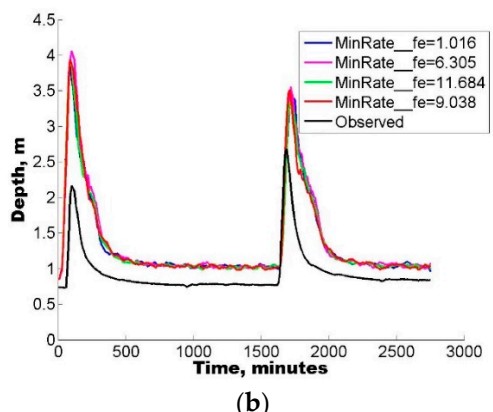

(**a**)　　　　　　　　　　　　　　　　　　　　　　　(**b**)

**Figure 19.** Depth levels of Sabinal river using (**a**) Nimperv parameter and (**b**) MinRate_fe parameter. The blue line is a minimum value, the magenta line is an average value, the green line is a maximum value, the red line is a parameter with maximum value of R2 and the black line are depth levels in hydrometric station.

## 4. Discussion

Some authors, such as Baek et al. and Sharifan et al. [42,43], suggest Nimperv and ManN as influential parameters in modeling with SWMM. Other researchers like Randall et al. [8] identified Horton's infiltration speeds as relatively sensitive parameters. Temprano et al. [44] also found the parameter with the highest sensitivity is the percentage of impervious surface. Guan et al. [45–47] report that the sensitive parameters in modeling in SWMM are those related to waterproofness, specifically in areas with depressions, Manning n of surfaces, and ducts.

This work implements the use of scarce data. Two different precipitation events and nine different input parameters were used. Each event was evaluated to show the robustness of the sensitivity analysis methodology in the hydrological modeling of urban basins, due to each individually analyzed parameter. The results show differences in the sensitivity of the model to the calculated parameters, where there is the accumulated depth as the comparison target.

For Event 1, it was found that the parameters with the highest sensitivity are Nimper and MinRate_fe. For Nimperv, cumulative depth values decrease while Nimperv values decrease and increase their values in the same way the parameter values grow. In the case of MinRate_fe, as the value of the parameter is lowered, the accumulated depth increases, and as the value of the parameter increases, the accumulated depth decreases. The above has a direct impact on the peak flow and runoff volume of the output hydrogram. For example, a high Nimperv value produces a dimmed output hydrogram with a higher runoff volume. A small Nimperv value produces a hydrogram with higher peak flow and lower runoff volume. In the case of MinRate_fe, a low parameter value produces an output hydrogram with high peak flow and higher runoff volume, and a high parameter value results in a hydrogram with a lower runoff volume and high peak flow.

In the case of validation with Event 2, the highest sensitivity parameter is Nimperv. The behavior of the parameter values and the values of the output hydrograms behave in the same way as in the previous case: its impact is on the volume of runoff and peak flow. MinRate_fe is less sensitive because the precipitation event is made up of two storms, and as a result, there is greater saturation in the ground at the end of the first peak of the storm. The remaining parameters for both Event 1 and Event 2 have a lower sensitivity relative to the result spectrum of the accumulated depths and have less influence on runoff volume and peak flow. In addition to the configuration of the basin and the location of the analysis point, the ManN parameter does not generate sensitivity in the model outputs. This may be due to the amount of base flow, the runoff generated by the nearby sub-basins, and the velocity.

According to the results, these parameters can be applied to carry out the calibration process. In turn, these parameters can be used individually or together. For example, in this case, $R^2$ efficiencies

greater than 0.60 were found, indicating that there is more than one possible Nimperv parameter value that calibrates the model with a reliable fit, that is, forming sets of two or more parameters and thus having a better fit to the observed hydrogram [48]. The results show that with the methodology used, it is possible to have reliable calibration with scarce data. According to this study, having more than one hydrometric station would be calibrated according to the area of influence of the station, applying the methodology used in this research.

For expressions for calculating the sensitivity of modeling results with respect to the reason for changing the parameters, it was found that they correctly identify the parameters that cause the most variation in the results. SWMM correctly represents the rainfall–runoff phenomenon, and the sensitivity of the input parameters depends on the characterization of the basin under study, precipitation and antecedent moisture, so the order of sensitive parameters is different for each area under study.

## 5. Conclusions

This study reflects the importance of sensitivity analysis of hydrological and hydraulic parameters interacting in a hydrological model of an urban basin, mainly because these parameters allow one to perform an adequate calibration of a model in which the runoff time series best fit the observed data obtained from hydrometry.

Therefore, an effort should be made to perform this type of analysis and give certainty to the results in modeling. Thus, it is also important to perform sensitivity analysis with different methodologies that fit the needs of the modeler, since each case study is different.

For the case study, it can be concluded that satisfactory results were obtained, achieving the objective of characterizing the sensitivity of the modeling parameters under a framework of hydrometric data shortages and obtaining the result that the most important parameters of this basin are Nimperv and MinRate. In addition, the analysis was not only performed using the Equations (4)–(6) but complemented by the calculation of the standard deviation, variance and $R^2$ of the result. Implemented $R^2$ shows that there are several parameters that provide a good representation of the system. The results show that the sensitivity of the parameters depends on the basin under study and the effects of secondary interactions between the model parameters. It is also shown that the most sensitive parameters in a simulation vary according to the storm and the accumulated precipitation. As Knighton [21] suggested, SWMM is well parameterized for the calculation of the rainfall–runoff ratio, so care must be taken when identifying sensitive parameters and the order in which they are applied.

In cases where data scarcity is high, the implementation of methods that enable the quantification of sensitivity in model predictions permit more reliable results.

Finally, subjectivity in rainfall–runoff modeling should be considered, since it depends mainly on the expertise decision-making of the modeler; in future studies, the uncertainty analysis of such models should be considered as well.

**Author Contributions:** H.A.B.-G. performed most of the analysis and numerical simulations shown in the study and also contributed to the manuscript preparation; V.H.A.-Y. contributed to the design of the study and discussion of the results; J.J.C.-R. revised the methodology, participated in the discussion of results and contributed to the final manuscript. R.S.-C. revised the methodology, participated in the discussion of results and contributed to the final manuscript. All authors have read and agreed to the published version of the manuscript.

**Funding:** This research received no external funding.

**Acknowledgments:** The authors express their gratitude to the Department of Urban Hydraulics of the Mexican Institute of Water Technology (IMTA) for their support in this investigation.

**Conflicts of Interest:** The authors declare no conflict of interest.

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
