# Peer review of "Sensitivity Analysis of the Rainfall–Runoff Modeling Parameters in Data-Scarce Urban Catchment"

_hydrology, doi:10.3390/hydrology7040073_

Round 1

Reviewer 1 Report

The authors answered and detailed my comments in a good way.
However is necessary to fix where appears: "Error! Reference source not found".

Author Response

Dear reviewer 1,

Reply to the observations:

  1. "Error! Reference source not found". This error was corrected. The cross references were placed again. This error was reviewed on computers with different versions of Word and the error did not appear. It was also converted to pdf and the document no longer displays this error. It is possible that in the process of submit the article the problem with the cross references was generated.

Reviewer 2 Report

This paper discusses sensitivity analysis (absolute-relative, relative-absolute and relative-relative sensitivity) of one hydraulic and eight hydrological parameters for an urban basin. The main conclusion of the paper is that the most important or sensitive parameters of this basin are Nimperv (Manning's N for impervious area) and MinRate (Minimum infiltration rate, mm/hr). The paper may be considered for publication after major revision.

Major comments:

Introduction section should clearly state the objectives and the research question to be addressed, this section fails to present a proper view of the problem and have insufficient references regarding the presented work.

Line 186, Methodology section is too short and should be more complete with a clear explanation. Sensitivity analysis could have been explained in a simpler way.

In result section, the statistical analysis in this paper looks to be incomplete and NSE or R2 must be include along with Mean µ, standard deviation σ, and coefficient of variation Cv.

The discussion in this section cannot extract useful information regarding the presented work. Sentence structure in the whole paper is not as much clear and easy to understand, the authors are advised to read whole paper very carefully and remove/change all the complicated sentences and grammatical mistakes.

The conclusion section does not summarize the complete work presented in the paper.

Minor comments:

The paper needs much attention in terms of grammar correction and sentence structure. The main problems are repetition of words, confused sentences, missing helping verbs (is, was, were) and use of pronouns (I, we, You) at some places.

  • Re-write lines from 32-34, 78-82, 88-89, with easy meaningful structure, I suggest to make correct sequences of phrases and then write with appropriate sentence structure.
  • Line 23, 44, 66, 107, 127, 135 [article is missing]
  • Line 71, [preposition mistake]
  • Line 94, [ grammatically incorrect]
  • Line 130, the singular verb is does not appear to agree with the plural subject basins
  • Line 114, figure 1 [spelling of hydrometric station is wrong]
  • Line 148, [missing word]
  • 140-141, [fonts used in the equation and their description are not identical].
  • Line 225-229 these sentences needs rewriting because it does not clearly explain the obtained result.
  • In some of the figures, the legends are too small to be legible.

Author Response

Dear reviewer 2,

Reply to the observations:

General comments were considered throughout the document.

  1. The Cross references are missing. This error was corrected. The cross references were placed again. This error was reviewed on computers with different versions of Word and the error did not appear. It was also converted to pdf and the document no longer displays this error. It is possible that in the process of submit the article the problem with the cross references was generated.
  2. Introduction section should state clearly the objectives and the research question to be addressed. The observation was modified in the document. Lines 90-103.
  3. Line 194-215, Methodology section is too short and does not clearly explained. The observation was modified in the document.
  4. In result section, the statistical analysis in this paper is unsuitable and NSE or R2 must be include along with Mean µ, standard deviation σ, and coefficient of variation Cv. observation was modified in the document. The calculation of R2 was added in the analysis. The observation was modified in the document. Lines 282-301, 355-357 and 363-372.
  5. Sentence structure in the whole paper is not as much clear and easy to understand, the authors are advised to read whole paper very carefully and remove/change all the complicated sentences and grammatical mistakes. The observation was modified in the document. The paper was sent to English editing.
  6. Re-write lines from 32-34, 78-82, 88-89, with easy meaningful structure, I suggest to make correct sequences of phrases and then write with appropriate sentence structure. The observation was modified in the document. Lines (33-34, 78-82, 88-89)

  1. The Line 114, figure 1 spelling of hydrometric station is wrong. The observation was modified in the document. Line 119.

  1. Line 148, sentence has some missing word. The observation was modified in the document. Line 152

  1. 140-141 fonts used in the equation and their description are not identical. The observation was modified in the document. Line 143-145.

  1. Line 225-229 these sentences needs rewriting because it does not clearly explain the obtained result. The observation was modified in the document. Lines 233-236.

  1. In some of the figures, the legends are too small to be legible. The observation was modified in the document, all the figures were review and add the information needed.

Reviewer 3 Report

The authors have used a sensitivity analysis to derive the rainfall-runoff calibration parameters most significant to modeling runoff in an urban basin drainage system. The main conclusion here seems to be that sensitivity analysis of input parameters is important. Yet that conclusion has been well established by the other papers cited (and certainly by other researchers using other models in other work).

My main concern with this paper is that I don’t see any real novelty here. It seems as if the study was conducted well and that the results were useful to understanding the runoff for the two events, but I don’t see how the paper delivers a new methodology or new understanding of the SWMM model. I recommended "major revision" because the text should be rewritten so that a clear statement is made as to what this paper adds to the literature on flood modeling, and then to support that statement point-by-point with the results. Some comments on specific sections follow.

Abstract. Specific mention of the rainfall-runoff ratio is made here and in the introduction, but it is not mentioned again until the conclusion section of the paper. How did this work contribute to the calculation of the rainfall-runoff ratio? If it didn’t contribute to it, why is it given such importance in the introduction of the work?

Introduction.

Line 90-96. With regard to this statement,

“This study presents a methodology for the analysis of local sensitivity with different expressions for its calculation, in the modelling of urban drainage, linking the changes in runoff with the variability of the parameters, as applied to the urban basin of the Sabinal River in Tuxtla Gutiérrez, Chiapas, Mexico. This work is a continuation of the sensitivity analysis carried out by [35] under the framework of data scarcity. This document addresses the sensitivity calculation, with three different equations using accumulated depths and two different precipitation events.”

what was done in your 2016 [35] work (and the work of others), and how does this work advance it? Are the three equations different from those typically or previously used? Possibly the main point of this work is to show that the most sensitive parameters in a simulation vary by storm and location (although in this study only the relative magnitude of the most sensitive parameters was different). The abstract states that the most sensitive parameters for the locations and events studied were the Manning coefficient of waterproof surface and the minimum infiltration speed. Was this a surprising result? The discussion suggests that others have found similar results.

English language usage. The language usage is good enough that I understood what was done, but there were many instances where word usage was not quite right. For publication, a copy edit by a native speaker would be a good idea. Here are a couple of examples.

Line 48. Just a language usage correction here: “…implemented a runoff model to determinate the flood…” should be “…implemented a runoff model to determine flood impact…”

Line 66-67. Another language correction here: “Therefore, the importance of performing a sensitivity analysis is letting know the set of parameters…” should be “Therefore, the importance of performing a sensitivity analysis is learning the set of parameters…”

In the Materials and Methods section, there are many instances of “Error! Reference source not found.” Please correct these.

Author Response

Reply to the observations made by the Reviewer 3:

  1. Line 90-96. How did this work contribute to the calculation of the rainfall-runoff ratio? The study includes the calculation of R2 which determines the correlation between the series of accumulated depths and those generated by the change of parameters; Therefore, reliable parameters are found that adjust the runoffs to the observed data. In the document graphs were added that show the behavior of the depths in the River with respect to the values of the parameters. Lines 90-96.
  2. what was done in your 2016 [35] work (and the work of others), and how does this work advance it? he abstract states that the most sensitive parameters for the locations and events studied were the Manning coefficient of waterproof surface and the minimum infiltration speed. Was this a surprising result? Regarding the sensitivity analysis in the previous work, a storm with a single peak was evaluated as input data and only a sensitivity index was calculated. In this work, a more robust sensitivity analysis was performed when evaluating with more than one equation. In addition, evaluating two different storms allows us to observe the behavior of the depths due to the parameters, under these conditions. In addition, some of the authors perform the sensitivity analysis by trial and error. Lines 90-103.
  3. Line 48. Just a language usage correction here: “…implemented a runoff model to determinate the flood…” should be “…implemented a runoff model to determine flood impact…” The observation was modified in the document. Line 48-49.
  4. Line 66-67. Another language correction here: “Therefore, the importance of performing a sensitivity analysis is letting know the set of parameters…” should be “Therefore, the importance of performing a sensitivity analysis is learning the set of parameters…” The observation was modified in the document. Line 68-69.
  5. In the Materials and Methods section, there are many instances of “Error! Reference source not found.” Please correct these. This error was corrected. The cross references were placed again. This error was reviewed on computers with different versions of Word and the error did not appear. It was also converted to pdf and the document no longer displays this error. It is possible that in the process of submit the article the problem with the cross references was generated.

Reviewer 4 Report

Summary:

The objective of the scientific work described in this manuscript is to present a sensitivity analysis methodology to determine critical parameters for using SWMM when monitoring data are scarce, as was the case with the urban catchment selected, to evaluate three different sensitivity indices. The results of the analysis support the importance and useful with data is scare in conducting sensitivity analysis to verify the most critical parameters. The manuscript also stresses the sensitivity of the parameters is linked to the specific catchment being characterized. Targeting monitoring resources to the critical parameters with a level of confidence allows for SWMM to be employed in conditions where it might not otherwise be possible due to resource limitations. 

Overall Comments:

The introduction did a thorough job of describing the use of sensitivity analysis in hydrologic modeling exercises. The final paragraph of the introduction could be improved with additional detail on the reasoning for the study and employing the three different equations. Perhaps this is covered in the previous work mentioned. In addition, the use of some phrases (such as "in relation to the above", "on the other hand", "therefore" and "however" were sometimes used too frequently or were just out of place and not necessary.

Unfortunately, the Figure and Table references in the copy were replaced with the error statements requiring me to determine which figure or table was being referenced. Although pretty straight forward it was cumbersome at times. The figures and tables, however, generally presented the data clearly. I also realize that the amount of data presented for the two events is significant and the decision was made to present each separately, but I found myself on subsequent reads wanting to compare the results side by side, especially the final sensitivity graphs for each event. Finally, in the discussion lines 357 - 361 describe how the hydrogram would respond to critical parameter increases/decreases. Presenting this as a hydrogram under the different scenarios might make this more impactful. 

Specific Comments:

Table 1 and 2: Suggest switching the last two columns so accumulated precipitation data is the last column.

Line 127/128: Change to Environmental Protection Agency

Line 138/139: Should read "In this study, infiltration loss is calculated..."

Line 203/204: Should read "The validation was also carried out with the Event 2,..."

Lines 211-212: Closed bracket missing after Event 1.

Line 226: Suggest being consistent and only using whisker and not moustache.

Line 230: Delete "According to the above" and just reference Table # (4?)

Line 246: Should be Figure 7, if I labeled the proceeding figures correctly and not Figure 1.

Line 260-269: Suggest highlighting the values that are discussed specifically in the associated text for easier reference as Table 5 contains a lot of data.

Figures 9 and 14: Should the units be %?

Line 294: Replace moustache with whisker.

Line 308-309: Delete sentence as it repeats the prior sentence.

Line 353-354: Consider revising sentence as it reads a little awkward.

Author Response

Dear reviewer 4,

Reply to the observations:

General comments were considered throughout the document. The last paragraph of the introduction was modified. In addition, the paper was sent to the MDPI English edition service, to improve English language and style. The cross references were placed again. This error was reviewed on computers with different versions of Word and the error did not appear. It was also converted to pdf and the document no longer displays this error. It is possible that in the process of submit the article the problem with the cross references was generated. The hydrographs were placed to understand their behavior with respect to the change of critical parameters.

  1. Line Table 1 and 2: Suggest switching the last two columns so accumulated precipitation data is the last column. The column was change to the end. Line 126-127.
  2. what Line 127/128: Change to Environmental Protection. The observation was modified in the document, text corrected. Line 130-131.
  3. Line 138/139: Should read "In this study, infiltration loss is calculated...". The observation was modified in the document. Text rewritten. Lines 141-142.
  4. Line 203/204: Should read "The validation was also carried out with the Event 2,...". The observation was modified in the document. Line 213-214.
  5. Lines 211-212: Closed bracket missing after Event 1. The observation was modified in the document. Line 220
  6. Line 226: Suggest being consistent and only using whisker and not moustache. The observation was modified in the document. Line 234.
  7. Line 230: Delete "According to the above" and just reference Table # (4?). The observation was modified in the document. Line 237.
  8. Line 246: Should be Figure 7, if I labeled the proceeding figures correctly and not Figure 1. The observation was modified in the document. Line 251.
  9. Line 260-269: Suggest highlighting the values that are discussed specifically in the associated text for easier reference as Table 5 contains a lot of data. The observation was modified in the document. The values were highlighted. Line 274-275.
  10. Figures 9 and 14: Should the units be %? The observation was modified in the document. Figures were change. Lines 279 and 361.
  11. Line 294: Replace moustache with whisker. The observation was modified in the document. Line 317.
  12. Line 308-309: Delete sentence as it repeats the prior sentence. The sentence was deleted.
  13. Line 353-354: Consider revising sentence as it reads a little awkward. The observation was modified in the document. Line 384-387.
  14. Line 286-300/363-371. Present an hydrogram under the different scenarios.

Round 2

Reviewer 2 Report

The paper may now be accepted for publication in the present form. All the observations raised are adequately addressed. 

Author Response

Dear reviewer 2,

Reply to the observations:

  1. English language and style are fine/minor spell check required. A complete revision of the document was made; to eliminate minor spelling errors.

Reviewer 3 Report

The authors have done a good job addressing the comments of the reviewers.  I have a much better idea now of what was done and what it adds to the body of knowledge in this area.  I have just a few more small items that I think it would be good for them to address:

Line 144 mentions y’, but the equation does not include y’. Please add the ‘ where it belongs in the equation.

For even better communication of the impact of your work, possibly restate lines 99-100 as you did in your reviewer response:

This work builds on our previous study [35] in which a storm with a single peak was evaluated as input data and a sensitivity index was calculated. In this work, a more robust sensitivity analysis is performed, as parameter influence is evaluated with more than one equation. In addition, we evaluate two different storms allowing us to observe the difference in behavior of the riverbed depths due to the parameters, under these differing conditions.

Also, why did you choose 9 parameters to evaluate and not 12 or 17 as others you cite did? It would be good to declare the reason for that choice as part of the above statement.

Author Response

Dear reviewer 3,

Reply to the observations:

  1. English language and style are fine/minor spell check required. A complete revision of the document was made; to eliminate minor spelling errors.
  2. Line 144 mentions y’, but the equation does not include y’. Please add the ‘ where it belongs in the equation. Added the ' in the equation. Line 149.
  3. Possibly restate lines 99-100 as you did in your reviewer response: The lines were repeated according to the previous response to the reviewer. According to the suggestion. It was also added in the same paragraph because the nine parameters used in the paper were chosen. Lines 99-105.

This manuscript is a resubmission of an earlier submission. The following is a list of the peer review reports and author responses from that submission.

Round 1

Reviewer 1 Report

Dear authors,

Below you can find my comments, suggests and corrects about the “Manuscript ID: hydrology 883744” submitted to Hydrology Journal.

Especific Comments:

Line 33 – “rainfall-runoff ratio” in this order the ratio value is greater than 1.0, maybe the inverse order is more adequate;

Line 93 – The Monte Carlo method is used for a stochastic process, the parameters sensibility is not a stochastic process, it’s just a scan search. Review this method denomination;

Figure 2 – Add map of Mexico with basin local indication. Also on the basin description is necessary to add area, concentrarion time, river slope, soil and use types;

Table 1 and 2 – Adopt the same decimal place number for all values in the same column;

Line 144 – The equation (1) was not used to reach the results, it’s a fraction of model method. Verify if the showing necessity;

Table 3 – The SWMM parameters have to be more detailed described, because they are the main concern of this manuscript;

Line 165 - The equations (2), (3) and (4) cannot have the same variable s(i,j) also is necessary show how is the criteria to obtain the variability range (deltas);

Line 178 – The node code information is irrelevant, suppress it

Table 4 – Use the same parameters ordering of the Table 3 as well the units for the values;

Table 5 – Instead of “Variance”, use “Coefficient of Variation” to be able to compare the results of each parameter with the others.;

Line 209 – The no variation in accumulated depth give us that the study area or the events are not appropriate for this reasearch;

General comments:

The manuscript results and conclusions obtained are just for the study area and for two events, and the study area has no representative, or it was not mentioned;

The unclear method describe puts difficult to replicate this research;

There are no result comparative with other authors results.

Reviewer 2 Report

Comments and Suggestions for Authors

Revision of paper titled: “Sensitivity analysis of the rainfall-runoff modelling parameters in data-scarce urban catchment” is required before considering for publication

General comments

I have read the submitted research article with interest and found that the topic “Sensitivity analysis of the rainfall-runoff modelling parameters in data-scarce urban catchment” is certainly of high importance particularly for the water managers in the data-scarce areas. The article describes the method of performing sensitivity analysis of hydraulic and hydrological parameters to further calibrate the Storm Water Management Model (SWMM) in the urban basin of Tuxtla, Gutiérrez, Chiapas, in Mexico. Two extreme precipitation events were selected to perform sensitivity and statistical analysis based on three approaches i.e. absolute-relative, relative-absolute and relative-relative.

However, while reading the manuscript I found it difficult to understand what the exact target of the study is. The writing style needs extensive improvement. The abstract is defining the model validation process in detail which is not required in the abstract. More details about study importance, catchment, model and variables used, period and results should be provided. The choice of method used is hardly justified and the structure and writing of the manuscript need to be improved throughout. As an example, the methods section jumps right into the method adopted, whereas to make it coherent with the general research article styles, it is suggested to start with materials (include study area details, and model or data used) and then define methods and analysis that has been performed. There is a lot of repetition of information as presented in figures and tables. The discussion of the results is totally missing. Figures (4 and 6) quality is not good enough and should be improved. 

From the manuscript as-is, it is not possible for me to fully judge the quality of the paper content-wise. Overall, I think the paper is not ready for publication in MDPI-Hydrology journal and it requires substantial editing and improved analysis of results.

 Specific comments:

  • SWMM is introduced directly in the abstract, so while using any abbreviation it is important to define the full form of abbreviation in its first place
  • L56-L60: the sentence is too much longer and should be split into three.
  • L62, “Some researchers, such as [17]…” here better to mention the name of Author of reference [17] i.e. Some researchers, such as Mannina G [17]….
  • Please, adjust references („such as [17], [18], [19]. …“) with author names, unless this is the format required by the journal.
  • L76 to L79: sentences are too long to understand the meaning. The objective of the study should be clear crisp.
  • L&9 to L83: same problem sentences are too much longer. Break them in shorter sentences to make clear the readability and understanding of the sentences. Specifically, you can start a new sentence from “this document will address…” in L81
  • In the Methodology section, you have mention the SWMM model second time in your article but without giving the full form of model “Storm Water Management Model (SWMM) ” in L88
  • L84: Heading Methodology: you have directly jumped into methodology adopted in this article and later you have given details of Study area and hydrological model and analysis used. It is suggested to start the 2nd section (Materials and Methods as the main heading followed by first with the study area, the model used, the method adopted and later analysis you performed in this paper. Accurate and logical study structure is important and suggested to make them improve the paper understanding and readability
  • L97 you are referring equations 2, 3 and 4 whereas these equations are not defined earlier to this reference. It is not logical to refer this way when something is coming after a few pages later
  • Whereas, equations 2, 3 and 4 are given after a few pages in section 3.3
  • L104-105 are empty and can be fixed
  • L112, you have mentioned that “Figure 2 shows the climatic seasons (yellow circles), as well as ….”, what do you mean by climatic season, rather you want to say climatic stations?
  • Be consistent in using terms climatic stations or Automatic stations as mentioned in Figure 2 legend
  • L124 and L125, you have given tables with precipitation events. For each column i.e. duration, ∆T, Accumulated precipitation etc. you have given units after “,”. It is suggested to mention units for each individual column in brackets i.e. Duration (min), ∆T (min), Accumulated precipitation (mm) coherent with international research articles style for tables with units
  • L124 and L125 you have given tables for precipitation characteristics from 7 climatic/ automatic station for two extreme precipitation events, whereas in Figure 2, you have mentioned 8 stations. Do not mention any station or information in maps/ figures which you are not using in your analysis i.e. station Solidaridad has not been mentioned in Tables 1 and 2
  • L136-L139, again sentence is too long. It is suggested to break this sentence into 2-3 short sentences i.e. one model introduction, model characteristics, its development, and applications etc.
  • L142, sentence “…physical processes mentioned above [23]” is not completely understood and it is suggested to give a brief explanation that how SWMM model calculates quantitative and qualitative assessments of runoff during precipitation events. Also, change reference style i.e. from “… physical processes mentioned above [23]” to “…physical processes mentioned above by James, W [23]”
  • L147, “…parameters are shown in Table 3, as well as…” no need to refer table in bold
  • L148, “…as well as the maximum and minimum values for modelling are observed…”, this sentence is not clear that maximum and minimum values of which parameter from the model are used to perform sensitivity analysis?. In my view, you want to say that the maximum and minimum values of hydrological and hydraulic parameters from the model are used to perform sensitivity analysis. Extensive rewriting is required to improve the paper quality to properly understand the paper objectives
  • L149, “…where the infiltration model used is Horton.”. Horton model used out of blue in this sentence and no further details are given anywhere in the article. Consider using only those methods, data and stations information which are explicitly used in this article or if you have adopted someone’s method, then refer it in a proper way
  • L150-L153, again sentence is too long and making it difficult to understand why and which parameters (as given in Table 3) are selected. The significance of the selected parameters and impacts on runoff generation process can be elaborated further
  • L166, make sure that expression ?? represents the m independent parameters of the model or this represents the jth independent parameters of the model
  • L168, “The expression (2) represents the relative absolute sensitivity, which describes the absolute change in the results for a relative change in parameters.”. I guess expression in equation 2 gives the absolute-relative sensitivity which might be mistakenly written the other way round. Please consider to correct it
  • L176, “The models were done with the input data …”, the non-scientific term “done” is used in this sentences, consider to replace it with some scientific term i.e. “The model's simulations/ run was carried out using input data …”
  • L181, “Figure 4a shows…” no need to refer Figure in bold
  • No need to refer a figure or its description when a figure is given a few pages after the reference as Figure 4a is refereed on page 6, whereas Figure 4 is given on page 9. Either consider use figure just before or after the description
  • Table 4 is the repetition of Figure 4a results. So no need to duplicate the information, use either figure or table to represent an information
  • L203, mention units of Mean, standard deviation and variances in Table 5
  • On lines L238, L248 and L252, Table 6, 7 and 8 are provided with the same information as given in Figure b-d. therefore, no need to give the same information twice. Use either tables or figures to give information
  • L228 to L237, you are concluding here that which model parameter is more sensitive based on standard deviations and variances from three adopted approaches. No need to conclude under the results section. So these sentences can be moved to either discussion or conclusion section
  • Tables 9, 10 and 11 can be combined is the repetition of information provided in Figure 4c, d and e. Consider removing the duplication
  • L268, “… Figure 5c (medium sensitivity) and Figure 5b…”, you have mentioned that Figure 5c, I guess it’s a typo mistake. Kindly consider to correct it to Figure 5a as given in Figure 5
  • L268-L273, unnecessary long sentence with no clear meaning that what author want to say here. Consider rewrite whole paper in short sentences with one verb per sentence. Avoid using too many information in a single sentence as this writing style is not understandable and also compromising the paper objective and results in the description in a logical way
  • L271-L272, no need to write about all parameters sensitivity. Only focus on the most sensitive parameters or the least one
  • L276- 277, “In the two figures mentioned above ….”, I guess the author wants to say “In the two figures mentioned below ….”
  • L276-L278, “In the two figures …. by 100 according to [24,25].” Is not clear
  • L81, Sub-heading 4.2, what does the author mean by “Event sensitivity analysis of 07/27/2011”? is it same as sensitivity analysis for the event 07/10/2011 but for a different precipitation event
  • L297 and L300, there are two spaces before the word “As” and words “Figure6a and word and”, kindly consider to check the spelling, grammar and format deeply
  • L307, Table 12 is the repetition of information as given in Figure 6a
  • L336, L346, L348, and L360 again these tables (Table 14, Table 15 and Table 16) are a repetition of Figure 6b-d. Delete the repetition
  • The author can combine information of statistical analysis of Tables 17, 18 and 19 can be combined in a single table
  • L386-L389, the sentence “This work raised the use of scarce data …. precipitation events, 07/10/2011 388 and 07/27/2011.” Seems a concluding sentence so can be moved to either discussion or conclusion section
  • Discussion of the results is totally missing in the existing draft. The author is suggested to highlight the reasoning on the selected most sensitive hydrological parameters that how they are sensitive to impact runoff generation process. How this work is better than existing work or justified in the study area. How this work can contribute to the existing work
  • L414 to L417, the sentence is too long and not clear
  • L423 to L426 is concluding the study findings. Again sentence is too much longer and not clearly reflecting on the concrete findings from this research

Reviewer 3 Report

This paper used Monte Carlo model to explore the sensitivity of SWMM on Sabinal River with the urban basin of Tuxtla, Gutiérrez, Chiapas, in Mexico. Overall the research is interesting and provides useful insights. However, there are several points that need to be addressed before publication in Hydrology:

  1. It is useful to show the subbasin of Sabinal River basin in a map(e.g., Figure 2).
  2. It need the basic input data of Sabinal River basin such as the area, average slope, width etc.
  3. The characteristics of rainfall event listed Table 1, 2. Could you also show hyetograph of rainfall event on Figure 3?
  4. Why are the sensitivity results of ManN the same in 1000 rainfall-runoff simulation?
  5. The sensitivity(Si) of  equation (2), (3), and (4) should expressed differently. 
  6. The authors should explain the physical meaning of the sensitivity value.
  7. The authors should explain whether the sensitivity results for each parameter are due to SWMM or watershed characteristics.